# Integration of ATAC-seq and RNA-seq Unravels Chromatin Accessibility during Sex Reversal in Orange-Spotted Grouper (*Epinephelus coioides*)

**DOI:** 10.3390/ijms21082800

**Published:** 2020-04-17

**Authors:** Xi Wu, Yang Yang, Chaoyue Zhong, Yin Guo, Tengyu Wei, Shuisheng Li, Haoran Lin, Xiaochun Liu

**Affiliations:** 1State Key Laboratory of Biocontrol, Guangdong Province Key Laboratory for Improved Variety Reproduction of Aquatic Economic Animals, Institute of Aquatic Economic Animals, School of Life Sciences, Sun Yat-Sen University, Guangzhou 510275, China; wuxi577@126.com (X.W.); guoyin10@163.com (Y.G.); lsslhr@mail.sysu.edu.cn (H.L.); 2Southern Laboratory of Ocean Science and Engineering, Zhuhai 519000, China

**Keywords:** chromatin accessibility, transcription factor, sex reversal, *Epinephelus coioides*

## Abstract

Chromatin structure plays a pivotal role in maintaining the precise regulation of gene expression. Accessible chromatin regions act as the binding sites of transcription factors (TFs) and cis-elements. Therefore, information from these open regions will enhance our understanding of the relationship between TF binding, chromatin status and the regulation of gene expression. We employed an assay for transposase-accessible chromatin with high-throughput sequencing (ATAC-seq) and RNA-seq analyses in the gonads of protogynous hermaphroditic orange-spotted groupers during sex reversal to profile open chromatin regions and TF binding sites. We focused on several crucial TFs, including ZNF263, SPIB, and KLF9, and analyzed the networks of TF-target genes. We identified numerous transcripts exhibiting sex-preferred expression among their target genes, along with their associated open chromatin regions. We then investigated the expression patterns of sex-related genes as well as the mRNA localization of certain genes during sex reversal. We found a set of sex-related genes that—upon further study—might be identified as the sex-specific or cell-specific marker genes that trigger sex reversal. Moreover, we discovered the core genes (*gnas*, *ccnb2*, and *cyp21a*) of several pathways related to sex reversal that provide the guideposts for future study.

## 1. Introduction

In teleosts, various patterns of sex determination, including gonochorism, protogynous hermaphroditism, and protandrous hermaphroditism, provide the abundant biological sources for academic and applied studies on sex determination/differentiation and sexual plasticity [1,2,3]. Groupers (*Epinephelus*) are protogynous hermaphroditic fish that undergo sex change from female to male during their life history [4]. They are considered good models for studying sex differentiation and sex reversal [5]. The orange-spotted grouper (*Epinephelus coioides*)—one of the many groupers—is an important economic species in Asia [4]. Under natural conditions, an orange-spotted grouper changes sex at approximately 4–5 years of age [6]. Therefore, to accelerate the process of sex reversal, exogenous androgens are used to induce sex reversal artificially, which has been widely used to explore the mechanism of sex reversal in groupers [5]. However, the molecular mechanism of sex reversal is still not well-understood. Previous studies reported that many TFs, such as the member of *sox* (sex determining region Y-box) [7,8,9], *wnt* (wingless/integrated) [10], *dmrt* (dsx and mab-3 related transcription factor) [11,12], and *nanos* [13], play important roles in the process of sex reversal. Yet, a comprehensive view of the transcriptional changes during sex reversal remains elusive.

Eukaryotic chromatin consists of repeating nucleosomes wrapped by short stretches of DNA and histones [14]. The position of nucleosomes provides the different accessibility of transcriptional machinery to cis-regulatory elements which is the DNA binding site or other regulatory motifs playing important roles in the regulation of gene transcription initiation, including promoters, enhancers, and silencers [15]. Furthermore, these cis-regulatory regions usually contain the binding sites of diverse TFs. Thus, the identification of cis-regulatory sequences in vivo is important for understanding how TF expression is coordinated throughout the grouper to facilitate sex reversal.

Chromatin immunoprecipitation-sequencing (ChIP-seq) is an ideal method to explore the interactions between in vivo DNA and protein; however, the lack of TF antibodies has limited the widespread application of this method in fish [16]. Consequently, development of a feasible and plastic method is crucial to identify the regulatory elements in fish genomes. To date, some new methods can be combined with the high-throughput sequencing to pinpoint potentially accessible regions of genome, such as DNase I sequencing (DNase-seq) [17], micrococcal nuclease sequencing (MNase-seq) [18] and formaldehyde-assisted isolation of regulatory elements sequencing [19]. ATAC-seq, a new method, was developed in 2013 and has been widely applied in many studies to detect the open chromatin regions. This technology takes advantage of the Tn5 transposase preloaded with sequencing adapters to probe the accessible open chromatin [20]. Thus, the process of ATAC-seq avoids multiple reactions and purifications which are required for library construction in sequencing. As a result, 5000 nuclei are sufficient for ATAC-seq which is 20- to 100-fold less than that required for MNase-seq or DNase-seq [21]. ATAC-seq has been applied in various species for different purposes combining multiple omics and technologies including ChIP-seq [22], fluorescence-activated nuclei sorting [23], single-cell RNA-seq [24], and RNA-seq [25].

To the best of our knowledge, our study is the first to use ATAC-seq with the crude nuclei of gonads during sex reversal in orange-spotted groupers. The usage of diverse gonads allowed us to unravel the differences of the chromatin accessibility among different developmental stages in order to explore the mechanism of sex reversal from a novel perspective. We correlated the atlas from ATAC-seq with RNA-seq to identify TFs networks and core genes in several pathways during sex reversal. In addition, a set of sex-related genes were also identified in the process.

## 2. Results

### 2.1. Artificial Sex ReverSal of Orange-Spotted Grouper

Initially, fish remained in the stage with abundant primary-growth oocyte (PO) and cortical-alveolus stage oocyte (PVO) before MT treatment (Figure 1a). Throughout the experiment, the gonads of fish in the control group also remained at the same stage (Figure 1b,d,f). On the contrary, fish in the MT-implanted group changed their sex from female to male. In histology, one week after MT implantation, oocytes in gonads degenerated, and new spermatogenic cysts proliferated, which was defined as the early stage of sex reversion (Week 1, Figure 1c). At three weeks after 17 alpha-methyltestosterone (MT) implantation, the gonads entered the middle stage of sex reversion, characterized by numerous spermatogonia (SG) and spermatocytes (SCs) and a limited number of oocytes (Week 3, Figure 1e). At five weeks after MT implantation, the gonads entered the late stage of sex reversion, possessing a majority of male germ cells, similar to the natural testes (Week 5; Figure 1g).

### 2.2. Proliferation Detection in the Gonad during Sex Reversal

To investigate the proliferation signal in the gonad during MT-induced sex reversal, the expression of proliferating cell nuclear antigen (PCNA) was examined using immunohistochemistry (IHC). In the Control group, the PCNA signals were mainly located in somatic cells rather than oocytes (Figure 2a). At the early stage of sex reversal (one week after MT-implantation), PCNA-positive signals were observed in the newly developed male germ cells (Figure 2b). At the middle stages of sex reversal (three weeks after MT-implantation), PCNA signals were detected mainly in the male germ cells (Figure 2c). At the late stages of sex reversal (five weeks after MT-implantation), the signals were only observed in SG and SC (Figure 2d). 

### 2.3. Apoptosis Detection in the Gonad during Sex Reversal

TdT-mediated dUTP nick end labeling (TUNEL) showed that the TUNEL signals can be hardly detected in the normal ovary (Figure 3a). At the early stage of sex reversal, TUNEL-positive signals could be found in some oocytes and somatic cells (Figure 3b). At the middle stage of sex reversal, TUNEL-positive signals were stronger and only observed in the oocytes (Figure 3c). At the late stage of sex reversal, the signals were in several cells which might be somatic cells or false positive signals (Figure 3d).

### 2.4. Application of ATAC-seq on the Gonads of Orange-Spotted Grouper

We integrated ATAC-seq and RNA-seq to investigate accessible chromatin, expression profiles of genes, and the relationship of them both during sex reversal in orange-spotted groupers. Orange-spotted groupers were divided into two groups—the MT-implantation group and the control group without MT-implantation. The gonadal histological characteristics during sex reversal were similar to those observed in our previous report [26]. The gonads of fish from three different stages, including the gonad filled with mostly primary-growth stage oocytes (Control group), the gonad after one week by MT-implantation (Intersex group), and the gonad after five weeks by MT-implantation (Testis group), were sampled in duplicate, respectively. Gonad cell nuclei of the gonad were incubated with Tn5 transposase combined with sequencing adapters to discern the open chromatin regions, but not the genomic regions that were tightly packed into nucleosomal arrays. The gonadal tissue homogenate of gonad was processed for RNA-seq (Figure 4a). We obtained an average of 73.34% mappability and 12.7 million qualified fragments per sample in ATAC-seq (Figure 4b). All libraries yielded fragment lengths with the expected distribution including a majority of small fragments (<200bp, representing internucleosomal open chromatin) and progressively fewer large fragments (≥200bp, representing the open chromatin spanning nucleosomes) (Figure 4c). The distribution of fragment lengths was similar to many previous ATAC-seq data [21,27]. Among the putative accessible regions detected by ATAC-seq, more than 50% were located in the 2 kb upstream of a transcriptional start site (TSS), especially the promoter in the 1 kb upstream of TSS, approximately 32% were located in distal intergenic regions, and the remaining 18% were located in intron, exon and downstream of gene bodies (Figure 4d). The accessible regions identified were mostly enriched within 2 kb of the TSS (Figure 4e), which is consistent with the presence of cis-regulatory elements (promoters, enhancers, silencers, etc.) in these regions in orange-spotted grouper. The detailed data in ATAC-seq and RNA-seq were displayed in Table 1 and Table 2.

### 2.5. Landscape of the Open Chromatin Regions in the Gonads during Sex Reversal

To evaluate whether the accessible regions were detected successfully as well as reproducibility of the replicates, strong enrichment of ATAC-seq peaks close to several sex-related genes was observed. The peak near *hsd17b7* (hydroxysteroid (17-beta) dehydrogenase 7) was significantly enriched in the Testis group, but was not detected in the Control group and the Intersex group (Figure 5a). PCA (Principal Components Analysis) was used to examine the relationships and reproducibility among biological replicates. PCA plots from the six ATAC-seq and RNA-seq datasets revealed a strong correlation between replicates of the same group and the correlation between the Control group and the Intersex group (Appendix A). From ATAC-seq, 4552 common peaks and numerous unique peaks were detected in the gonads of three developmental stages (Figure 5b). In RNA-seq, 534 common genes were detected among the comparison of six samples (Figure 5c). There were 1291 up-regulated genes and 7 down-regulated genes in the Control group compared to the Intersex group. A total of 2888 up-regulated genes and 326 down-regulated genes were identified in the Control group compared to the Testis group. The Testis group had 984 up-regulated genes and 843 down-regulated genes compared to the Intersex group (Figure 5d). 

### 2.6. Highly Enriched Known Motifs and Their Target Genes in Differentially Accessible Regions

To identify potential drivers of the differences in chromatin accessibility, we screened out the enrichment of the top ten up-regulated TF motifs in differentially accessible regions when they compared to those within constitutively open regions. The enrichment and DNA sequences of the top ten motifs in the Control group versus the Intersex group were listed (Figure 6a). Among them, the most enriched motifs were ZNF263 (zinc finger protein 263), SPIB (transcription factor Spi-B), and RFX4 (regulatory factor × 4). The DNA sequences of ZNF263 were overlapped with the first 21 nt ZNF263 motif identified by ChIP [28]. A relatively clear footprint of TF occupancy was detected near the aggregated ZNF263 full sites in accessible chromatin regions (Figure 6b). The other enriched TFs motifs, such as SPIB, KLF9 (krueppel-like factor 9), and ETV2 (ETS translocation variant 2), showed similar footprints in the genome (Appendix A). In the comparison of the Control group and the Testis group, the top ten up-regulated and down-regulated known TF motifs were listed in Appendix A. RFX4, RFX2, and RFX3 were the most enriched up-regulated motifs, and KLF9, ZNF263, and SP2 were the most enriched down-regulated motifs. When the Intersex group was compared to the Testis group, SP3, SP1 (specificity protein 1), and KLF9 were the most enriched up-regulated motifs, and RFX4, RFX3, and RFX2 were the most enriched down-regulated motifs (Appendix A).

Among the target genes of the most enriched motifs, numerous genes were related to sex reversal, which formed into TF-gene networks. Inspection of the top enriched TF-bound gene lists showed that many genes were involved in cell proliferation, cell differentiation, and steroid synthesis, including *zbtb16* (zinc finger and BTB domain containing 16), *dmrt1*, *sox11*, *star* (steroidogenic acute regulatory protein), and *hsd17b7* [8,11,29,30,31]. Several genes, including *smad4* (mothers against decapentaplegic homolog 4) and *zbtb16*, were common target genes among different TFs. To illustrate the potential regulatory network, we plotted the networks of ZNF263, SPIB, ETV2, and their target genes in the comparisons of the Control group and the Intersex group (Figure 6c). 

### 2.7. Expression Levels of Sex-Related Differentially Expressed Genes (DEGs) in ATAC-seq and RNA-seq Data

Using a sequence analysis tool—MEME—we found the known motifs of TFs enriched in a highly sex-preferential manner combining the FPKM (fragments per kilo base millions) values of the genes encoding TFs from RNA-seq data (Figure 7). The detailed enrichment and expression of the thirty DEGs related to sex reversal were shown in Appendix A. Notably, these thirty DEGs are essential for gonadal development and sex regulation, including *bmp15* (bone morphogenetic protein 15), *cyp19a1* (Cytochrome P-450AROM), *dazl* (deleted in azoospermia-like), *dnd* (dead end), *dmrt1-3*, *foxl2* (forkhead box protein L2-like), *nanos2-3*, *piwil1* (piwi-like protein 1), and *sox9*. Furthermore, the timing of the motifs coming out was consistent with the expression of their corresponding genes. For example, both the binding motifs and the expression levels of *dazl* and *dmrt2* were strongly enriched in gonads after five weeks of MT-implantation. A similar observation was made for *cyp19a1*, *hsd17b7*, *klf4* (krueppel-like factor 4), and *sox9*. Interestingly, certain sex-related genes stood out depending on the gene expression patterns in the RNA-seq. For instance, the expression levels of *dnd*, *nanog*, *nanos3*, and *sall4* (sal-like protein 4) were significantly higher in the ovaries and intersexual gonads—female preference, and the expression levels of *dmrt1*, *gas8* (growth arrest-specific protein 8), *piwil1*, and *star* were significantly higher in testis—male preference. 

### 2.8. Validation of DEGs Inferred from ATAC-Seq and RNA-seq Data

Among thirty DEGs, RT-qPCR of twelve genes was conducted to validate the credibility of the transcriptome data and profile the expression patterns of the genes during sex reversal. The results showed that the expression patterns of these genes were largely consistent with the transcriptome data. The expression of *bmp15*, *dnd* (dead end), *gdf9* (growth/differentiation factor 9), and *nanog* in ovary was significantly higher than in testis, and their expression decreased gradually during sex reversal (Figure 8a,d,e,g). The expression of *dmrt1*, *nr5a2* (nuclear receptor subfamily 5 group A member 2), *star* (steroidogenic acute regulatory protein), *sox9* (sex determining region Y-box 9), and *rergl* (ras-related and estrogen-regulated growth inhibitor-like gene) in testis were dramatically higher than in ovary, especially in the gonad of five weeks after MT-implantation (Figure 8b,f,j,k,l). 

### 2.9. Localization of Sex-Related Genes

The localization of sex-related genes in different gonadal stages was analyzed by in situ hybridization (ISH). The primers used for ISH are listed in Table 3. The *dazl* mRNA was restricted to both female and male germ cells (oocytes, SG, and SC) in orange-spotted grouper (Figure 9a,b). Interestingly, the *dazl* mRNA concentrated on the Balbiani body as a perinuclear speckle in early primary-growth stage oocytes, after which the speckle disappeared and ultimately distributed uniformly in late-primary-growth stage oocytes. This expression pattern was similar to the distribution of *Ecslbp2* in primary-growth stage oocytes [26]. *Dmrt1* signals couldn’t be detected in the ovary of natural females (Figure 9d). *Dmrt1* signals were not detectable in the ovaries from the natural females (Figure 9d). Gonads at the late stage of sex reversion had abundant SG, SC, and ST with *dmrt1*-positive signals and oocytes without *dmrt1* signal (Figure 9e). As a rate-limited enzyme in the pathway to synthesize steroid hormone, *star* was specifically expressed in the testes of male orange-spotted groupers [30]. No *star* signal was detected in oocytes (Figure 9g). Furthermore, it was found that *star* mRNA expressed in SG, SC, and ST (Figure 9h). Moreover, its expression levels in three male germ cells decreased gradually. ISH results showed that *rergl* mRNA was only detected in male germ cells (Figure 9j,k), which is consistent with the RT-qPCR results. Sense riboprobes of *dazl*, *dmrt1*, *star*, and *rergl*—as control experiments—showed no signal in the gonads (Figure 9c,f,I,l).

### 2.10. Core Peaks of ATAC-seq Inferred from RNA-seq in the Early Stage of Sex Reversal

Correlation of ATAC-seq and RNA-seq was analyzed in order to explore the changes in transcription activity and capture the crucial TFs during the early stage of sex reversal. In RNA-seq, the top 20 Kyoto Encyclopedia of Genes and Genomics (KEGG) enriched pathways were listed in Figure 10a, such as the lysine degradation pathway, regulation of actin cytoskeleton pathway, and so on. In the estrogen signaling pathway, there were 17 DEGs in the Control group versus the Intersex group, e.g., *gnas* (guanine nucleotide-binding protein G(s) subunit alpha), *grb2* (growth factor receptor-binding protein 2), *esr2* (estrogen receptor beta), etc. Among these genes, we only found the peak of *gnas*, which is upstream of the pathway in the same comparison from ATAC-seq result. In regard to oocyte meiosis, 13 DEGs were detected, including PKA (protein kinase A), SMC1 (structural maintenance of chromosome 1), and PGR (progesterone receptor), etc.; however, only *ccnb2* (G2/mitotic-specific cyclin-B2) peak was discerned in the ATAC-seq data. In view of the steroid hormone biosynthesis pathway, several enzymes, such as HSD17B1 (17-Beta-Hydroxysteroid Dehydrogenase Type 1) and CYP19A involved with the biosynthesis of steroid hormone had taken significantly changes in the process of artificial sex reversal. Most of the DEGs were lower than the expression in the Control group except CYP21A (steroid 21-monooxygenase) which was exactly found in ATAC-seq (Figure 10b). The maps of the estrogen signaling pathway, oocyte meiosis pathway, and steroid hormone biosynthesis pathway were displayed in Appendix A.

## 3. Discussion

In this study, we used an artificial method to induce sex reversal by MT-implantation. During the process, the oocytes underwent apoptosis, whereas the male germ cells exhibited proliferation. We used the sensitive ATAC-seq method to locate the regions of open chromatin during sex reversal. The data were used to unveil key drivers, define a set of sex-related genes in different stages of gonads, and attempt to determine the transcriptional networks during sex reversal in orange-spotted grouper.

When comparing differentially enriched TFs, certain core TFs, e.g., ZNF263, SPIB, ETV2, and KLF9 regulated more downstream TFs and genes; therefore, they constitute regulatory networks that play roles in sex reversal. The ZNF263 is a member of zinc finger proteins containing a C2H2 zinc finger domain and a KRAB repression domain which is associated with basic cellular processes, including development, differentiation, metabolism, apoptosis, and cancer [32,33,34]. However, few studies have identified the role of ZNF263 with regard to reproduction; in contrast, our results indicated that ZNF263 called on plenty of sex-related genes during sex reversal. This result implied that ZNF263 might play important roles in sex reversal through the transcriptional regulation of its targets. KLF9 (krüppel-like transcriptional factor 9), a member of the KLF family containing the C2H2 zinc finger domain, too, was initially identified as a transcriptional repressor in the liver of rat [35]. Subsequently, it was found that KLF9 promotes the differentiation of adipocyte in 3T3-L1 cells and glioma stem cell death [36,37]. However, the actual role of KLF9 remains unclear and requires further study in orange-spotted grouper during sex reversal.

By integrating ATAC-seq data with RNA-seq data from different developmental stages, we were able to characterize certain open chromatin regions and genes in these developmental stages. Importantly, we preliminarily defined a set of sex-related genes with high expression levels in one or the more developmental stages. Furthermore, the expression profiles and characterization of partially sex-related genes (*dazl*, *dmrt1*, *star*, and *rergl*) were validated during sex reversal. ISH results suggested that *dazl* is a germ cell marker with high expression in testis which was consistent with the expression of *dazl* in other species, such as Asia seabass [38], Nile tilapia [39], and medaka [40]. It has been reported that *dmrt1* expressed specifically in spermatogenic cells (SG, SC, and ST) [11]. Our results indicated that *dmrt1* was only detected in male germ cells and could therefore be considered as a germ cell marker in orange-spotted grouper. Previous study has shown that *star* is expressed specifically in testis [30], meanwhile, our results provided the exact location of *star* in the testis of orange-spotted grouper. Based on our result, *rergl* is a male-specific gene without expression in the ovary. Prior to our current study, there was no reports about the sexual dimorphism of *rergl* that implied its potential role in sex regulation. However, unlike other genes, *cyp17a1* exhibited an entirely different expression pattern in ovary and testis and its distribution varied in different germ cells [41]. The condition of *cyp17a1* hinted at the distinct roles in female and male which is agreed with the results in zebrafish [42]. These data will provide useful guideposts to distinguish different cell types; however, the specific functions of these sex-related genes in gonadal development requires further study.

Combining the ATAC-seq and RNA-seq results, we found that certain accessible chromatin regions interact with some core genes during the early stage of sex reversal. In the estrogen signaling pathway, G-protein coupled estrogen receptor 1 (GPER) receives the signal of estrogen, then GPER promotes the expression of *gnas*, accordingly, *gnas* increases the activity of adenylate cyclase 1 which affects cyclic AMP-responsive element-binding protein 1 [43]. Eventually, the target genes are initiated to take part in the cell cycle, apoptosis, and cell adhesion, as well as in components of the membrane and cytoplasmic signaling cascades. In ATAC-seq, the peak of *gnas* was regulated by ZNF263, which suggested that changes in chromatin of *gnas* might be a core gene triggering the differential expression of downstream genes in the pathway. In regard to oocyte meiosis, progesterone activates a series of related genes, and *ccnb2* is one of the bottom genes that prevents DNA replication and meiosis I progression and cell proliferation. Several upstream changes accumulate gradually and result in the changes in chromatin accessibility near *ccnb2*. In view of the steroid hormone biosynthesis pathway, CYP21A, the crucial enzymes for transforming steroid to cortisol [44], were found in ATAC-seq regulated by a TF—SP1. In teleosts, cortisol is the primary glucocorticoid regulating a series of physiological processes, such as metabolism, immunity, and growth [45]. Besides, cortisol was reported a peaking expression during sex change in anemonefish [46] and promoted the masculinization in Pejerrey [47], which suggests a role in sex reversal. Our results illustrated that the synthesis of estrogen and androgen is inhibited, while at the same time the synthesis of cortisone is up-regulated during artificially induced sex reversal, providing novel insight regarding sex reversal in orange-spotted grouper.

In summary, we herein provided a novel resource for identifying open chromatin regions in the gonads at different developmental stages. Our dataset was established based on the sex reversal by artificial MT-implantation in protogynous hermaphroditic orange-spotted grouper. We integrated ATAC-seq data with RNA-seq results and anticipated that our ATAC-seq data will be useful for integration with further genomic analyses as well as other epigenetic information in orange-spotted grouper. Such studies are important for understanding the complex networks of TFs and their target genes, exploring the potential mechanism of sex reversal, and identifying a set of sex-related genes for distinguishing sex and cell types.

## 4. Materials and Methods

### 4.1. Animals

Orange-spotted groupers were raised in the Guangdong Daya Bay Fishery Development Center (Huizhou 516,081, Guangdong, China). The fish were kept in indoor pools under controlled water temperatures of 22.7–27.8 °C. All fish were anesthetized with MS222 for 15 min, then sacrificed. All animal experiments were conducted in accordance with the guidelines and approval of the respective Animal Research and Ethics Committees of Sun Yat-Sen University 

### 4.2. MT-Induced Sex Reversal

Sex reversal of orange-spotted groupers was induced by MT (Sigma-Aldrich, St. Louis, MO, USA) treatment referred to our previous paper with minor modification [26]. All Fish (body weight, 1.90 ± 0.65 kg; body length, 43.75 ± 9.25 cm) were divided into two groups, control group (*n* = 25) and MT implantation group (*n* = 25). The dosage of MT was 10 mg/kg body weight, and the prepared strips were implanted in the abdomen of fish. Before implantation (Week 0), five fish gonads were obtained randomly to confirm developmental stage. After MT implantation, the gonadal tissues of five fish were sampled randomly each week from two groups, respectively. The whole experiment was conducted for 5 weeks (Table 4). For each fish, the gonad was cut into three parts for different purposes, including histological examination, ISH, and RNA extraction. 

### 4.3. Proliferation and Apoptosis Assays

IHC analysis was performed using the modified method as previously described [48]. Briefly, the frozen gonadal sections were rehydrated in PBS for 30 min at room temperature, blocked with 5% BSA (Sigma-Aldrich) in PBS, and then incubated with the first antibody of PCNA (1:100 diluted with 2% BSA in PBS) (Sigma-Aldrich) overnight at 4 °C. After washing several times by PBS, the sections were blocked with 5% goat serum in PBS for 1 h, and incubated for 1 h at room temperature with the second antibody HRP-conjugated anti-mouse IgG (Boster, Wuhan, China) at a dilution of 1:2000 with 2% goat serum in PBS. Signals were detected by the TSA^TM^ Plus Fluorescence System (Roche, Mannheim, Germany). Cell nuclei were stained by 4’, 6-diamidino-2-phenylindole (DAPI, blue). 

Apoptosis in the gonad was examined by a TUNEL assay with a TUNEL Bright Green Apoptosis Detection kit (Vazyme, Nanjing, China) following the manufacturer’s instructions. Briefly, the frozen sections were rehydrated in PBS for 15 min at room temperature, incubated with 100 uL 20 ug/mL Proteinase K solution for 10 min, and then incubated with TdT buffer for 10–30 min at room temperature. At last, cell nuclei were stained by DAPI. Sections processed without terminal deoxynucleotidyl transferase were used as a negative control. Photographs of the samples were taken under a light microscope (TCS SP5, Leica, Wetzlar, Germany).

### 4.4. Histology Analysis

After fixation in Bouin’s solution for 24 h, gonads were completely embedded in paraffin. The solidified paraffin blocks were sectioned and stained with hematoxylin and eosin (H&E). After mounting, the gonadal sections were observed by Nikon light microscopy (Tokyo, Japan).

### 4.5. ATAC-Seq

Nucleic suspensions were incubated in a transposition mix that included a transposase. The transposase entered the nuclei and preferentially fragmented the DNA in open regions of the chromatin. Simultaneously, adapter sequences were added to the ends of the DNA fragments. Following the transposition reaction at 37 °C for 30 min, the products were purified with a MiniElute DNA Kit (Qiagen, Hilden, Germany), amplified as described in a previous study [21], and sequenced using an Illumina HiSeq^TM^ 4000 by Gene Denovo Biotechnology Company (Guangzhou, China).

### 4.6. Read Alignment

Bowtie2 (version 2.2.8, Baltimore, MD, USA) [27] with the parameters “–X 2000” was used to align the clean reads from each sample against the reference genome (unpublished data), and reads aligned to the mitochondria were filtered depending on the mitochondria genome of orange-spotted grouper [49]. For all data files, duplicates were removed using Picard (Broad Institute/MIT, Cambridge. MA. USA). 

### 4.7. Peak Scanning

All reads aligning to the + strand were offset by +4 bps, and all reads aligning to the – strand were offset by −5 bps. Shifted, concordantly aligned paired mates were used for peak calling by MACS (version 2.1.2, Boston, MA, USA) [50] with parameters “--nomodel --shift -100 --extsize 200 -B -q 0.05”. MACS is a computational method that was designed to identify read-enriched regions from sequencing data. Dynamic Possion distribution was used to calculate *P* value of the specific region based on the uniquely mapped reads. Q values were calculated from *P* values using Benjamini-Hochberg procedure [51]. The region would be defined as a peak when Q value < 0.05.

### 4.8. Peak-Related Genes Annotation

According to the genomic location information and gene annotation information of peak, peak related genes can be confirmed using ChIPseeker (version v1.16.1, Guangzhou, China) [52]. In addition, the distribution of peaks in different genomic regions (such as promoter, 5′UTR, 3′UTR, exon, intron, downstream regions, and intergenic regions) was assessed.

### 4.9. Irreproducible Discovery Rate

To measure the consistency between biological replicates within an experiment, we used the irreproducible discovery rate (IDR, version 2.0.2, Berkeley, CA, USA) [53] to evaluate the reproducibility of high-throughput experiments with parameters “--input-file-type narrowPeak --plot”. IDR calculations using scripts provided by the ENCODE project (https://www.encodeproject.org/software/idr/) were performed on all pairs of replicates using an oracle peak list called from merged replicates for each condition, keeping only reproducible peaks showing an IDR value of 0.05 or less. 

### 4.10. Analysis of Motif and TF Footprints

The DNA binding sites for specific TFs or histone modifications which showed conserved DNA sequence patterns were not random. MEME suite (http://meme-suite.org/) was used to detect the motifs. We used MEME-ChIP to scan motifs with high reliability through peak regions, and used MEME-AME to confirm the existence of any specific known motifs.

If TF complexes bind DNA sequences, the steric hindrance will appear when the Tn5 transposase is used to dispose of the genome. Thus, the ATAC-seq reads can be used to reflect the TF footprints [21].

### 4.11. Differential Analysis of Multi-Samples

DiffBind (version 2.8.0, Cambridge Institute University of Cambridge, UK) was used to analyze peak differences across groups. We identified significantly differential peaks with FDR < 0.05 in two comparison groups. Using the same method, genes associated with different peaks were annotated, and enrichment analysis of Gene Ontology (GO) functions and KEGG pathways were identified.

### 4.12. RNA-Seq

Gonads were thawed for chloroform extraction and evaluated for RNA quality and concentration. Libraries were prepared using Nugen Ovation RNA-seq System V2 and Nugen Ovation Ultralow Library System kits and sequenced on an Illumina HiSeq2000 (Illumina, San Diego, CA, USA). 

### 4.13. DEGs and Enrichment Analysis

For each sequenced library, the read counts were adjusted by the edgeR program package by one scaling normalized factor (Bioconductor, Roswell Park Cancer Institute, Buffalo, NY, USA). Differential expression analysis was performed using the DEGseq R package (1.20.0). The Benjamini and Hochberg method was used to adjust the *P* values. Corrected *P*-value of 0.05 and log 2 (fold change) of 1 were set as the threshold for significantly differential expression.

GO enrichment analysis of DEGs was implemented by the GOseq R package. GO terms with a corrected *P* value less than 0.05 were considered significantly enriched by DEGs. In addition, KOBAS software was used to test the statistical enrichment of DEGs in KEGG pathways.

### 4.14. Validation of DEGs from transcriptome data by RT-qPCR

The relative mRNA levels of twelve DEGs (*bmp15*, *dmrt1*, *dmrt2*, *dnd*, *gdf9*, *nr5a2*, *nanog*, *nanos2*, *hsd17b7*, *rergl*, *star*, and *sox9*) from transcriptome data were examined by RT-qPCR to validate their expression profiles in the gonad of the Control group and one week after MT implantation, respectively. Another main purpose of analyzing the expression of these genes by RT-qPCR is to determine whether their expression profiles paralleled those inferred from RNA-seq data. 

Using the First Strand cDNA Synthesis Kit (Roche), 1 µg total RNA was extracted by TRIzol® (Invitrogen, Carlsbad, CA, USA), then was reverse transcribed with random primers. According to the manufacturer’s instruction, the reverse transcription process was performed at 25 °C for 10 min, 55 °C for 30 min, 85 °C for 5 min, and 4 °C for 5 min. The RT-qPCR reaction was performed using the SYBR Green PCR master mix (Roche). In a 10 μl reaction volume, the amplification regime was conducted with 40 cycles of 94 °C for 20 s, 57 °C for 10 s, and 72 °C for 20 s, and followed by amplification at 72 °C for 5 min. *Ef1a* was used as a reference gene for sample normalization. The specific primers used in this study are listed in Table 5.

### 4.15. ISH Analysis of Several Sex-Related Genes in Gonads

The ISH protocol for several sex-differed genes (*dazl*, *dmrt1*, *rergl*, and *star*) was described in our previous study [26]. Briefly, after fixing with 4% paraformaldehyde (diluted with 0.1% DEPC water) at 4 °C overnight, fresh gonad samples were dehydrated in 30% sucrose at 4 °C overnight. Then the samples were embedded by an optimal cutting temperature compound (SAKURA Tissue-Tek®, Atlanta, GA, USA). Gonad blocks were cryosectioned at 5–6 μm, then the sections were mounted on a Superfrost™ Plus microscope slides (Thermo Fisher Scientific, Waltham, MA, USA) especially for ISH. The different cDNA fragments of the five genes were inserted into the pGEM-T Easy vector to synthesize the sense and anti-sense riboprobes labeled by digoxigenin (DIG, Roche). The hybridization buffer consisted of 0.5–1 μg/mL DIG probes, 50% deionized formamide, 5 × SSC (saline sodium citrate), 0.5 mg/mL salmon sperm RNA, 1 × Denhart’s solution, and 5% dextran sulphate. After dropping 200–300 μL hybridization buffer, each slide was covered with a 50 × 50 μm coverslip in a sealed box and incubated overnight in an oven at 65 °C. The slides were washed in SSC and PBS (phosphate buffer saline) buffer several times. Later, sections were blocked with Blocking Reagent (Roche) for 1 h. The DIG label was tested with an alkaline phosphatase-conjugated anti-DIG antibody (Roche, diluted 1:1000) for 2 h and stained with NBT/BCIP Stock Solution (Roche). 

### 4.16. Statistical Analysis

All data were expressed as mean values ± SEM. Significant differences were checked by one-way analysis of variance (ANOVA) and Student’s t-test was used, and a probability level less than 0.05 (*p* < 0.05) was used to indicate significance. All data were performed and analyzed by GraphPad Prism 5.0 (GraphPad Software, San Diego, CA, USA).

### 4.17. Data Availability

In present study, ATAC-seq data and RNA-seq data can be obtained from the Transcriptome Shotgun Assembly project DDBJ under accession number PRJDB9134.

## 5. Conclusions

To the best of our knowledge, our study presents the first open chromatin analysis in orange-spotted grouper during sex reversal using the high-sensitive ATAC-seq method. This technique allowed the precise mapping of potential gene regulatory regions. By integrating ATAC-seq analysis with RNA-seq data, we identified numerous novel sex-related genes for different developmental stages. Further mining of the open chromatin regions identified by ATAC-seq revealed the networks of important TFs with downstream genes, as well as the changes of chromatin accessibility that might play important roles in sex reversal. This study provided important datasets and multiple novel avenues of investigation for the future study.

## Figures and Tables

**Figure 1 ijms-21-02800-f001:**
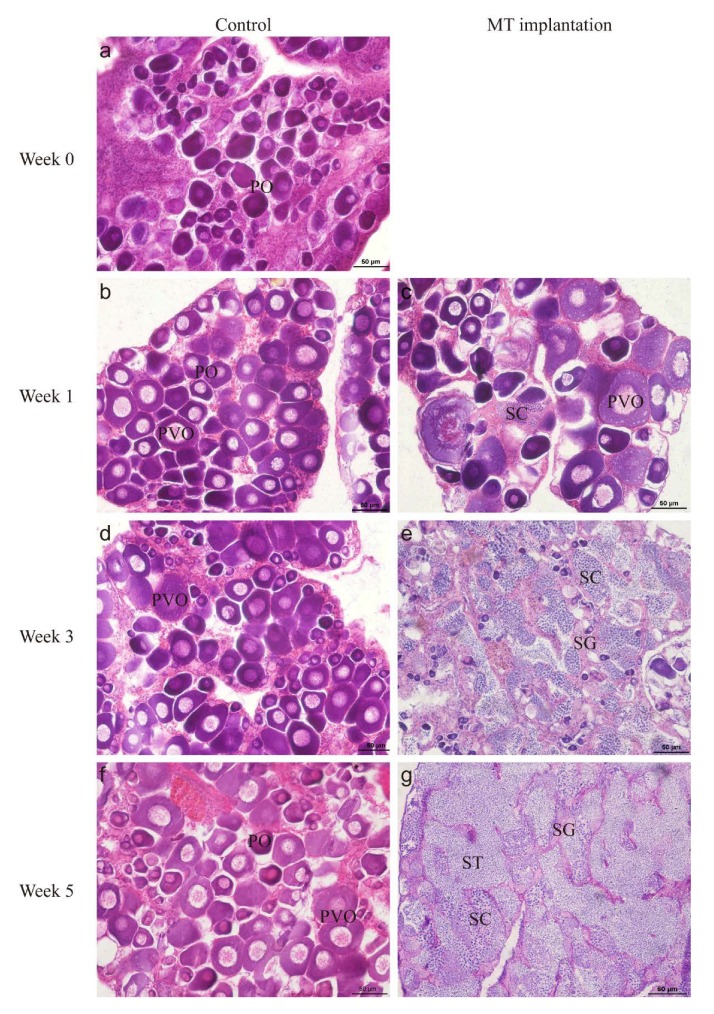
Gonadal histological morphology of MT treatment on gonads of the orange-spotted grouper. (**a**,**b**,**d**,**f**) Histology of gonads in control fish. (**c**,**e**,**g**) Histology of gonads after MT implantation. PO, primary-growth stage oocyte; PVO, cortical-alveolus stage oocyte; SG, spermatogonia; SC, spermatocytes; ST, spermatids. Scale bars = 50 μm.

**Figure 2 ijms-21-02800-f002:**
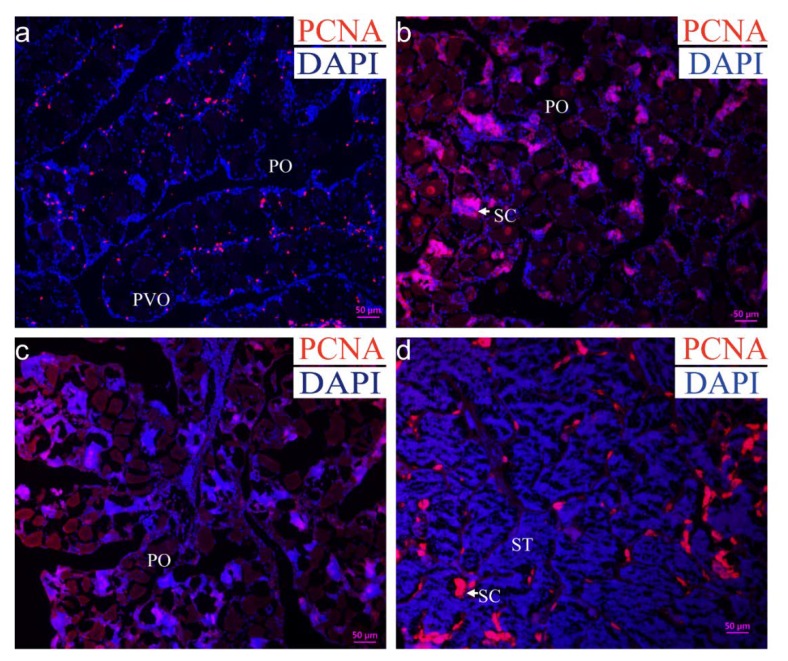
The proliferation detection in the gonad during sex reversal. IHC was performed on the gonad in different stages with antibody against PCNA (red). The sections were counterstained with DAPI (blue), and the merged signals are purple. (**a**) Ovary in control group with the portion of mostly PO, (**b**) gonad after one week by MT-implantation (the early stage of sex reversal), (**c**) gonad after three weeks by MT-implantation (the middle stage of sex reversal), (**d**) gonad after five weeks by MT-implantation (the late stage of sex reversal). PO, primary-growth stage oocyte; PVO, cortical-alveolus stage oocyte; SC, spermatocyte, ST, spermatid. Scale bars = 50 μm.

**Figure 3 ijms-21-02800-f003:**
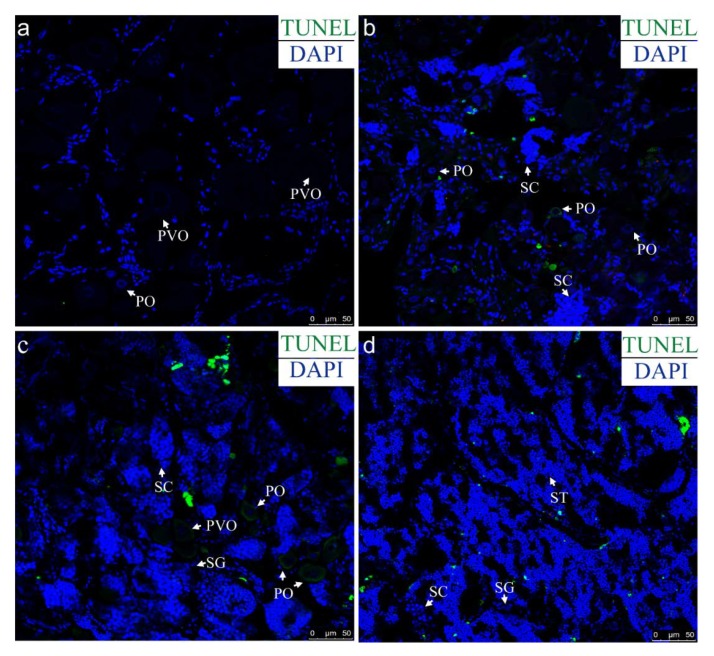
TUNEL assay in the gonads during sex reversal. TUNEL signals were green, and the sections were counterstained with DAPI (blue). (**a**) Ovary in sham group with the portion of mostly primary-growth stage oocytes, (**b**) gonad after one week by MT-implantation, (**c**) gonad after three weeks by MT-implantation, (**d**) gonad after five weeks by MT-implantation. PO, primary-growth stage oocyte; PVO, cortical-alveolus stage oocyte; SG, spermatogonia, SC, spermatocyte, ST, spermatid. Scale bars = 50 μm.

**Figure 4 ijms-21-02800-f004:**
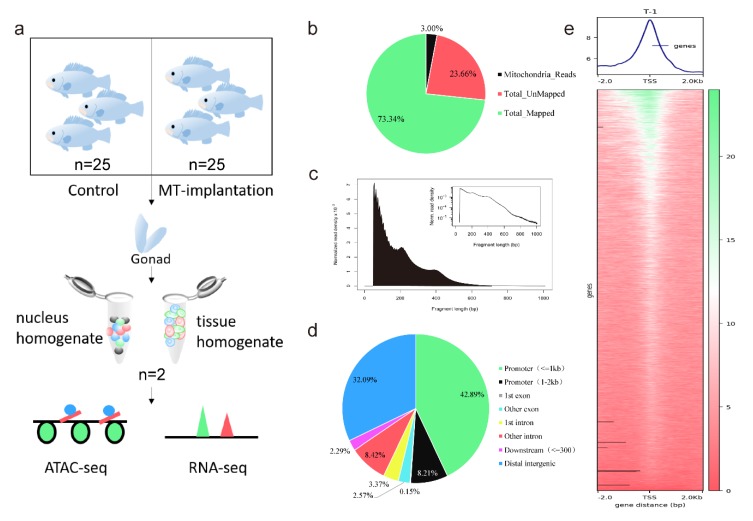
Overview of ATAC-seq results in the gonad of orange-spotted grouper. (**a**) Experimental workflow. Gonads from two groups (Control group and MT-implantation group) were sampled, then cell nuclei homogenate and tissue homogenate were extracted and processed for ATAC-seq and RNA-seq, respectively. (**b**) Average genomic distributions of mitochondria identified in ATAC-seq datasets. (**c**) Fragment lengths within a representative ATAC-seq library. The small fragments represent sequence reads in open chromatin (<150 bp), the peak at ~200 bp results from sequence reads that span one nucleosome, and larger peaks represent progressively more compact chromatin. (**d**) Proportions of the ATAC-seq peak regions that represent the various genome annotations, compared to the representation of a given sequence element in the orange-spotted grouper genome. (**e**) In a representative sample (T, Testis group), enrichment of ATAC-seq signals around the TSS. Top, the aggregated enrichment plot around all TSSs. Bottom, the enrichment heatmap near an individual TSS. TSS, transcriptional start site.

**Figure 5 ijms-21-02800-f005:**
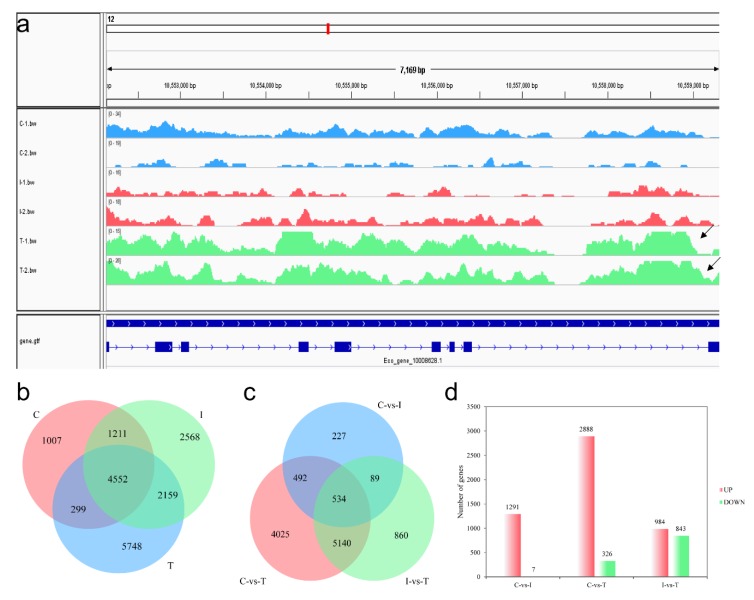
ATAC-seq and RNA-seq profiles. (**a**) ATAC-seq tracks for *hsd17b7* located on chromosome 12 in six biological replicates. The putative open regions were labeled with black arrows. (**b**) Venn diagram of the peaks in different groups from ATAC-seq analysis. (**c**) Venn diagram of the genes in different comparison groups from RNA-seq analysis. (**d**) Histogram of up-regulated and down-regulated genes in different comparison groups from RNA-seq analysis. C, Control group; I, Intersex group; T, Testis group.

**Figure 6 ijms-21-02800-f006:**
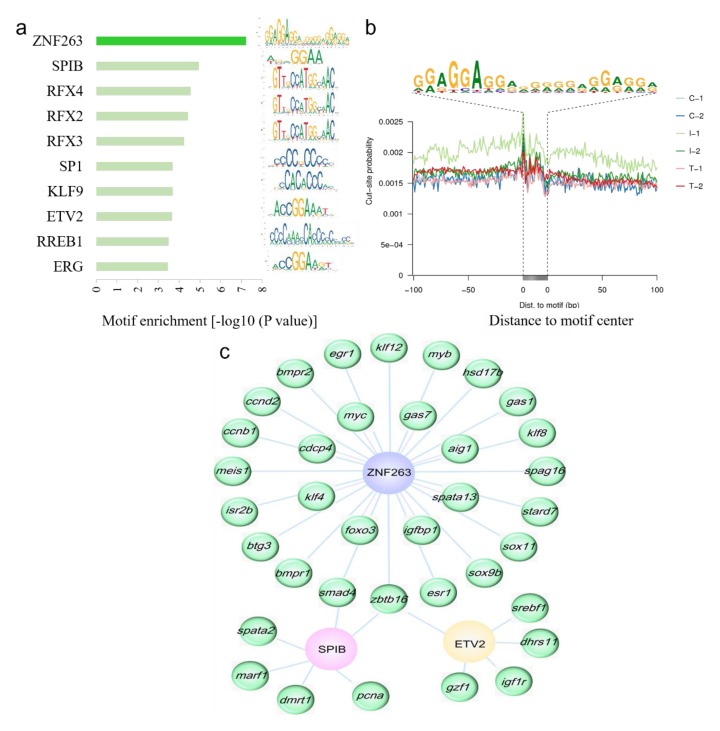
The most enriched motifs and their target genes. (**a**) The top ten known enriched motifs in the comparison between the Control group and the Intersex group. The right motifs were the corresponding DNA sequences of the TFs. (**b**) ATAC-seq footprint at the ZNF263 full site. Cut sites probability, insertions per site are normalized to have the same average depth of insertions ± 100 bp away from motif center. The DNA sequence of the bottom is the motif of ZNF263. C, Control group; I, Intersex group; T, Testis group. (**c**) Transcriptional regulatory network in the three enriched TFs in the comparison between the Control group and the Intersex group. ZNF263, SPIB, and ETV2 are TFs, the rest are target genes.

**Figure 7 ijms-21-02800-f007:**
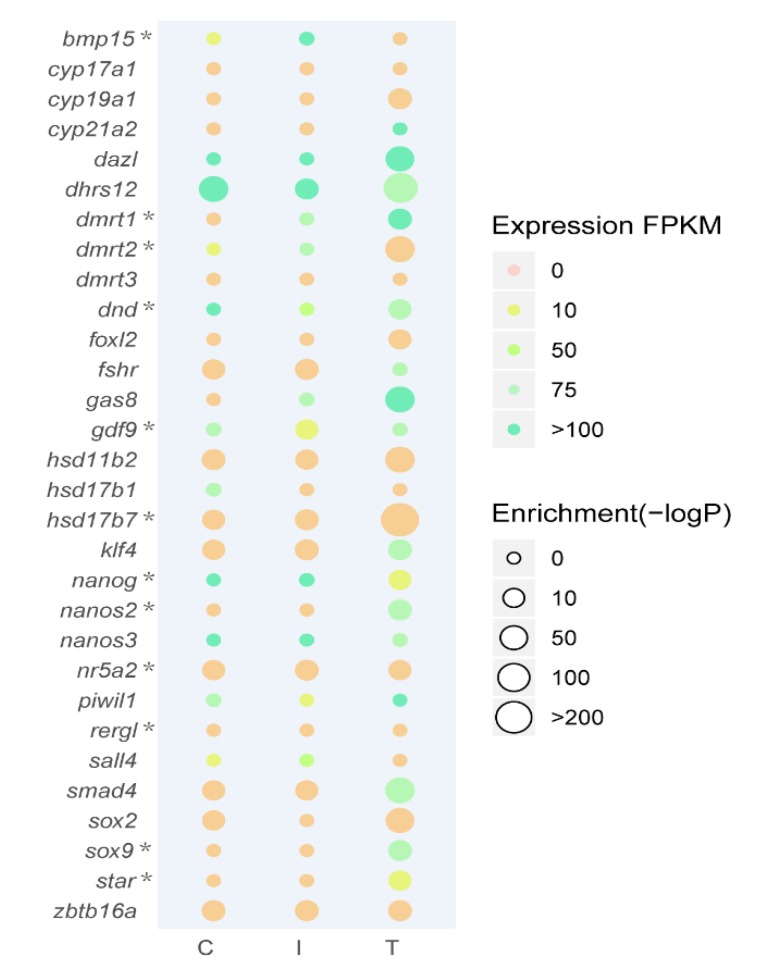
Bubble diagram based on the expression levels of thirty DEGs from RNA-seq and the motif enrichment from ATAC-seq. The color of the bubble represented the FPKM value in RNA-seq, and the size of the bubble represented the enrichment in ATAC-seq. The gene names were labeled on the left, and the genes validated by real-time PCR (RT-qPCR) were marked with asterisks after the gene names. C, Control group; I, Intersex group; T, Testis group.

**Figure 8 ijms-21-02800-f008:**
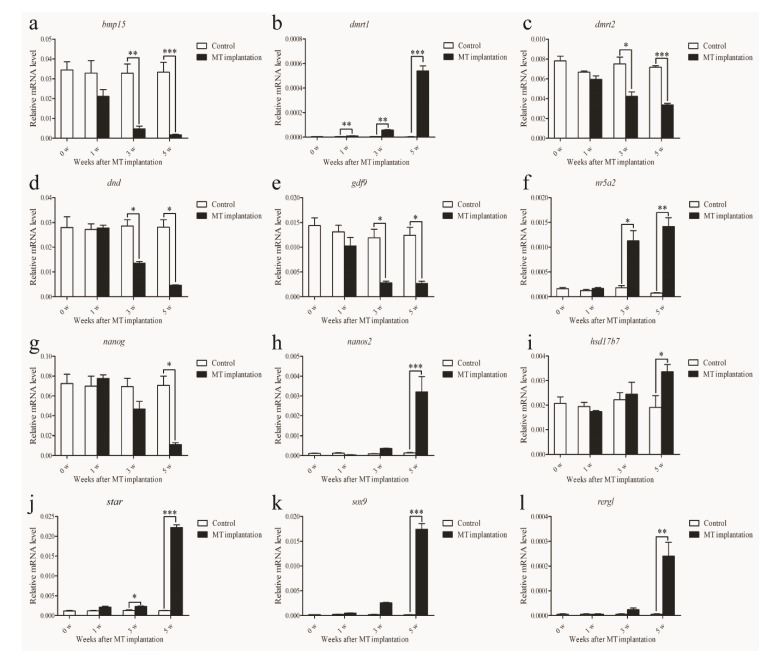
RT-qPCR validated the expression of the selected genes during sex reversal. 0 w, the female grouper before MT implantation; 1 w, the early stage of sex reversal after 1week implantation; 3 w, the middle stage of sex reversal after 3 weeks implantation; 5 w, the late stage of sex reversal after 5 weeks MT implantation. The data was shown as mean ± SEM (*n* = 4 or 5) and the values with different numbers of asterisks were significantly different with a probability level < 0.05 (*p* < 0.05). * *p* < 0.05, ** *p* < 0.01, and *** *p* < 0.001 between the Control group and MT-implantation group. *Ef1a* was used as the reference gene. (**a**) *bmp15*; (**b**) *dmrt1*; (**c**) *dmrt2*; (**d**) *dnd*; (**e**) *gdf9*; (**f**) *nr5a2*; (**g**) *nanog*; (**h**) *nanos2*; (**i**) *hsd17b7*; (**j**) *star*; (**k**) *sox9*; (**l**) *rergl*.

**Figure 9 ijms-21-02800-f009:**
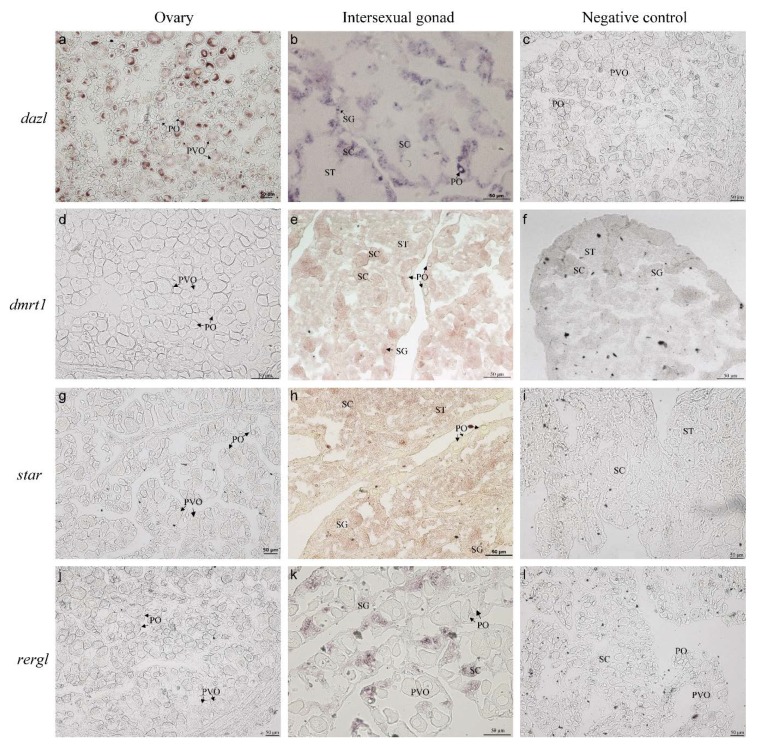
The cellular location of four sex-related genes from different gonads. Gene names were provided on the left side. (**a**,**d**,**g**,**j**) Sections of the ovary without any treatment. (**b**,**e**,**h**,**k**) Sections of the intersexual gonad. (**c**,**f**,**i**,**l**) Negative contrast hybridized by the sense probes. PO, primary-growth stage oocyte; PVO, cortical-alveolus stage oocytes; SG, spermatogonium; SC, spermatocyte; ST, spermatid; SZ, spermatozoa. Scale bars = 50 μm.

**Figure 10 ijms-21-02800-f010:**
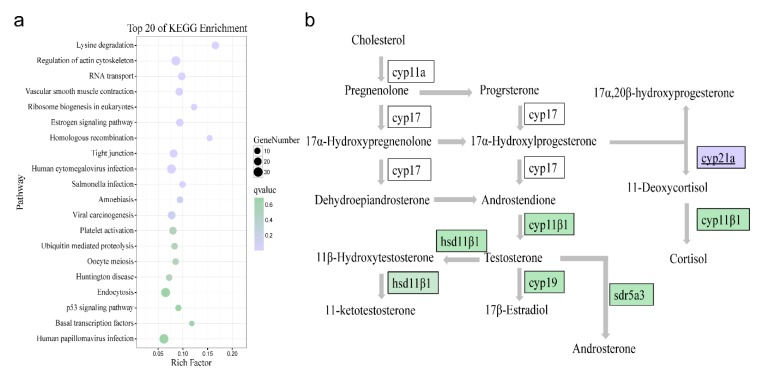
Core peaks in the important pathway. (**a**) Top 20 pathways of KEGG enrichment in the comparison of the Control group and the Intersex group from RNA-seq. (**b**) Sketch of the sex steroid biosynthesis pathway. The purple rectangle represented up-regulated genes and the green rectangle represented down-regulated genes in the gonad of the early stage of sex reversal. Underlined genes mean the gene was detected via ATAC-seq.

**Table 1 ijms-21-02800-t001:** Summary of the ATAC-seq data.

Sample	Total Reads	Mitochondrial Reads	Clean Reads	Total Mapped	Clean
C-1	158,840,360	3.93%	96.07%	77.22%	89,605,877
C-2	179,216,998	1.81%	98.19%	71.16%	93,719,129
I-1	123,712,724	5.84%	94.16%	74.27%	66,091,411
I-2	120,042,218	4.81%	95.19%	76.06%	65,930,437
T-1	59,402,610	0.65%	99.35%	79.69%	37,601,624
T-2	14,693,8476	0.54%	99.46%	78.36%	86,143,916
Total	788,153,386	-	-	-	439,092,394

Mitochondrial Reads, the ratio of the total reads mapping to the mitochondria genome. Clean Reads, the ratio of the reads filtering out the mitochondrial genome. Total Mapped, the ratio of the total reads mapping to the reference genome. Clean, the number of the reads after filtering. C, Control group; I, Intersex group; T, Testis group.

**Table 2 ijms-21-02800-t002:** Summary of the RNA-seq data.

Sample	Raw Data	Clean Data (%)	Mapped Reads (%)	Total Mapped (%)
C-1	38,082,536	38,005,438 (99.80%)	31,366 (0.08%)	92.43%
C-2	37,104,826	37,030,360 (99.80%)	45,746 (0.12%)	91.76%
I-1	39,789,128	39,709,566 (99.80%)	36,968 (0.09%)	92.24%
I-2	41,232,406	41,154,704 (99.81%)	9006 (0.02%)	93.12%
T-1	39,531,884	39,445,640 (99.78%)	135,722 (0.34%)	90.93%
T-2	39,575,790	39,496,980 (99.80%)	117,852 (0.30%)	91.56%
Total	235,316,570	-	-	-

Clean Data, the number and ratio of high-quality reads from the raw data. Mapped Reads, the number and ratio of the reads mapping to the ribosome. Total Mapped, the ratio of the reads mapping to the reference genome. C, Control group; I, Intersex group; T, Testis group.

**Table 3 ijms-21-02800-t003:** The primers used for ISH.

Primers	Sequence (from 5′ to 3′)
*dazl*-ISH-F	CAACCAGACTTCACCTTTCC
*dazl*-ISH-R	AGTGAGGTGGAGGGTACTG
*dmrt1*-ISH-F	GCTGGTAGTTGTACAGGTT
*dmrt1*-ISH-R	GACCACCAGATCTCCTTT
*star*-ISH-F	CAACTTTCAAGCTGTGCGCT
*star*-ISH-R	GACCAAGGGACCTCGTTAGC
*rergl*-ISH-F	TTGGGACTGTCCAACCACTT
*rergl*-ISH-R	GTTCGCAGATGGCAACTCAT

**Table 4 ijms-21-02800-t004:** The experimental design of artificial MT implantation.

	Number	Control Group	MT Implantation Group
Time			
0 week		5	0
1 week		5	5
2 weeks		5	5
3 weeks		5	5
4 weeks		5	5
5 weeks		5	5

**Table 5 ijms-21-02800-t005:** Primers used to verify the quality of samples.

Primers	Purpose	Sequence (from 5′ to 3′)
*ef1a*-F	RT-qPCR	GGTCGTCACCTTCGCTCCAT
*ef1a*-R	RT-qPCR	TCCCTTGGGTGGGTCATTCT
*bmp15*-F	RT-qPCR	GTGAGCCTCATCTTCAAGTC
*bmp15*-R	RT-qPCR	TCAGAACATCCAGTGACGTA
*dmrt1*-F	RT-qPCR	GGCTATGTGTCTCCTCTGAA
*dmrt1*-R	RT-qPCR	ATTCTTCACCATCACCTCAG
*dmrt2*-F	RT-qPCR	AAGCTTTCCATGAAACTCAA
*dmrt2*-R	RT-qPCR	AGAAAGTCTTTGCCGTACCT
*dnd*-F	RT-qPCR	GGTGGAGAGAGTGTCTCTGA
*dnd*-R	RT-qPCR	GGTTTGTTTGAAGACAGTGG
*gdf9*-F	RT-qPCR	GTCTCCTCCTCTGCTTCTTT
*gdf9*-R	RT-qPCR	GACTTTTATCGCCTCGTTTA
*nr5a2*-F	RT-qPCR	GTACCAGTACACAGCCTTCC
*nr5a2*-R	RT-qPCR	ATCTTATTCTGCACCACCAC
*nanog*-F	RT-qPCR	AGACTGGAAGACGCAGATAA
*nanog*-R	RT-qPCR	ACTCCTCATGAGTCTTGTCG
*nanos2*-F	RT-qPCR	GACTACTTCACCCAGGAACA
*nanos2*-R	RT-qPCR	TCTGACTTCAGGTTGTGTGA
*hsd17b7*-F	RT-qPCR	GGTCTGTACTCATCCGTCAT
*hsd17b7*-R	RT-qPCR	GATTCAGGCTTTTGCTTAAA
*rergl*-F	RT-qPCR	CACCTAATCAGAGAGCTCCA
*rergl*-R	RT-qPCR	GAAGAATATGAGTGCCACCT
*star*-F	RT-qPCR	ATCTTCAGGCACTTTCTCAA
*star*-R	RT-qPCR	TGATGTTTTCAGAGGAGCTT
*sox9*-F	RT-qPCR	GTTCGGACACTGAGAACACT
*sox9*-R	RT-qPCR	GACGTGAGGTTTGCTTTTAC

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
