# Peer review of "Integration of ATAC-seq and RNA-seq Unravels Chromatin Accessibility during Sex Reversal in Orange-Spotted Grouper (Epinephelus coioides)"

_ijms, 2020, doi:10.3390/ijms21082800_

Round 1
Reviewer 1 Report
The manuscript by Wu et al. is focused on an interesting topic: expression of genes during sex reversal. The authors induced sex reversal via androgen treatment and performed two techniques enabling insights into the expression changes. To me, the data are useful (although sample size is rather limited), the approach is sound and the topic interesting. However, the authors should be more careful in writing and presentation of their results:
- Plural/singular are often mixed, typos are present - e.g. „entichment“ within Figure 3a, „annalysized“ on page 8 etc., some words are out of place e.g. „concerted“ on page 10, „vital peaks“ sound strange as well, many misuse of „while“ and other conjunctions). I am not an English native speaker myself, but I strongly feel that language should be improved considerably (it is mostly but not always understandable, but far from perfect).
- More biological/zoological background is urgently needed, the authors did not even define their studied organism well, they state only that the fish are „groupers“ without scientific name of the taxon. Please, supplement the Introduction with much more information in this respect.
- Be careful in explaining abbreviation at the first appearance in the text, e.g. „MT“ and „TSS“ are without any explanation on the second page, „T group“ is not well explained, „ISH“ in the legend of the Table 3 as well etc.
- Figure 1: there were 3 treatment groups, not 2 as one could infer from the figure 1a.
- Please, define a term and then use it consistently. Sometimes you use rather vague terms and change them throughout the manuscript, for example „genes with sex-biased expressions“ are subsequently assigned as „sexual preference“ (page 6, not at all fitting term), „sex-differed genes“ (very awkward), „sex-related genes“, „sexual dimorphism“…
- Use correct order of links to particular figure parts, „Figure. 5e, a, d, and g“ looks strange, similarly further. Also, do not write a dot after „Figure“, it is not an abbreviation.
- Use the same font in the figure and in the legend of it (e.g. Figure 6a versus 6A in the figure itself).
- The first two paragraphs of Discussion are not informative and should be deleted.
- Lines 295-297: the sentence does not make much sense.
- Line 363: Please, be explicit what were the minor modifications, it might be important. Also, please describe the handling of animals and procedures with them in more details, that is important.
- Line 373: A crucial reference genome is assigned as „unpublished data“.
- Line 382: „Q value lower than 0.05“ – please, explain.
- Line 432: The way of normalization using ef1a is described incorrectly in this sentence.
Author Response
Dear reviewer,
Thank you for your comments concerning our manuscript entitled “Integration of ATAC-seq and RNA-seq unravels the chromatin accessibility during sex reversal in orange-spotted grouper (Epinephelus coioides)” (Manuscript Number: ijms-747820). Those comments are all valuable and very helpful for revising and improving our paper, as well as the important guiding significance to our research. We have studied comments carefully and have made correction that we hope meet with approval. Revised portion are marked in red in the paper. Regarding the questions (Q) raised by these reviewers, we would like to answer (A) them as follows.
The manuscript by Wu et al. is focused on an interesting topic: expression of genes during sex reversal. The authors induced sex reversal via androgen treatment and performed two techniques enabling insights into the expression changes. To me, the data are useful (although sample size is rather limited), the approach is sound and the topic interesting. However, the authors should be more careful in writing and presentation of their results:
A: We thank this reviewer’s complimentary remarks.
Q1. Plural/singular are often mixed, typos are present - e.g. „entichment“ within Figure 3a, „annalysized“ on page 8 etc., some words are out of place e.g. „concerted“ on page 10, „vital peaks“ sound strange as well, many misuse of „while“ and other conjunctions). I am not an English native speaker myself, but I strongly feel that language should be improved considerably (it is mostly but not always understandable, but far from perfect).
A1. Thanks for your kind advice. We have in fact gone back to rewrite many parts of the manuscript in order to improve its grammar, syntax and readability. I hope that the revised manuscript will represent a better version in terms of the English language. And we have corrected the mistakes and the misused words in the revised manuscript (Line 21, 22, 42, 52, 70, 130, 164, 212, 220, 247, 255, 290, 295, 306, 318, 328, 344, 368, 397, and so on).
Q2. More biological/zoological background is urgently needed, the authors did not even define their studied organism well, they state only that the fish are „groupers“ without scientific name of the taxon. Please, supplement the Introduction with much more information in this respect.
A2. Thanks for your kind suggestion. We have added more information about our experimental animal—orange-spotted grouper in the revised manuscript (Line 39-43).
Q3. Be careful in explaining abbreviation at the first appearance in the text, e.g. „MT“ and „TSS“ are without any explanation on the second page, „T group“ is not well explained, „ISH“ in the legend of the Table 3 as well etc.
A3. Thanks for your kind suggestion. We have provided the full names for the first occurrence of abbreviation and explained the special words, such as “MT”, “TSS”, “ISH”, and many gene names. (Line 44-46, 79, 84, 85, 128, 143, 167,171, 182-184, 198, and 201-203).
Q4. Figure 1: there were 3 treatment groups, not 2 as one could infer from the figure 1a.
A4. Thanks for your question. All Fish (body weight, 1.90±0.65kg; body length, 43.75±9.25cm) were divided into two groups, control group (n=25) and MT implantation group (n=25). Before implantation (Week 0), five fish gonads were obtained randomly to confirm the developmental stage. After MT implantation, the gonadal tissues of five fish were sampled randomly every week from two groups, respectively. The whole experiment was conducted for 5 weeks. So, the number of fish in figure 1a cannot represent the real number in the experiment. We are sorry for the inconvenience in reading, and we have added the detailed explanation in the “Materials and Methods” (Line 366-375) and labeled the real number in figure 1a. After verifying the developmental stage of the gonad by paraffin section, we selected three stages during sex reversal (the gonad with the portion of mostly primary-growth stage oocytes, the gonad with a few of male germ cells, and the gonad with a mass of male germ cells) with two replicates, respectively, to conduct ATAC-seq and RNA-seq.
Q5. Please, define a term and then use it consistently. Sometimes you use rather vague terms and change them throughout the manuscript, for example „genes with sex-biased expressions“ are subsequently assigned as „sexual preference“ (page 6, not at all fitting term), „sex-differed genes“ (very awkward), „sex-related genes“, „sexual dimorphism“…
A5. Thanks for your kind suggestion. We have corrected the vague terms into “sex-related” in the revised manuscript (Line 26, 27, 75, 142, 196, 207, 237, 238, 266, 297, 305, 314, 316, 329, 357, 483, and 508).
Q6. Use correct order of links to particular figure parts, „Figure. 5e, a, d, and g“ looks strange, similarly further. Also, do not write a dot after „Figure“, it is not an abbreviation.
A6. Thanks for your kind suggestion. We have changed the order of links and the corresponding context (Line 196-203),and we have deleted all the dots after “figure” (Line 79, 80, 83, 86, 87, 90, 105, 106, 109, 114, 115, 119, 145, 148, 149, 150, 154, 156, 166, 170, 172, 186, 188, 199, 213, 224, 228, 230, 241, 245, 247, 249, 251, 253, 266, 275, 287, 288, and 290) in the revised manuscript.
Q7.Use the same font in the figure and in the legend of it (e.g. Figure 6a versus 6A in the figure itself).
A7. Thanks for your kind advice. We have changed the font in figure 6 according to the figure legend.
Q8. The first two paragraphs of Discussion are not informative and should be deleted.
A8. Thanks for your kind advice. We have simplified the first two paragraphs of Discussion into two sentences in the revised manuscript (Line 295-298).
Q9. Lines 295-297: the sentence does not make much sense.
A9. Thanks for your kind advice. We have deleted the sentence in the revised manuscript.
Q10. Line 363: Please, be explicit what were the minor modifications, it might be important. Also, please describe the handling of animals and procedures with them in more details, that is important.
A10. Thanks for your kind advice. In our first implantation, we prepared 120 fishes and spent 7 weeks to collect samples. The histology analysis showed that the gonads in the same group developed synchronously, and almost all fish became male after 5 weeks. Therefore, in this experiment, we shortened the implantation time to 5 weeks and decreased the amount of fish to lower the cost and time. That’s our “minor modifications” we didn’t explain clearly. Besides, we have added more description about the handling of animals and procedures in the revised manuscript (Line 366-381).
Q11. Line 373: A crucial reference genome is assigned as „unpublished data“.
A11. Thanks for your kind advice. The genome of orange-spotted grouper, a job of our laboratory, has done and the related paper has submitted. We believed that it will be published soon.
Q12. Line 382: „Q value lower than 0.05“ – please, explain.
A12. Thanks for your question. MACS (version2.1.2) is a computational method that was designed to identify read-enriched regions from sequencing data. Based on the unique mapped reads, dynamic possion distribution was used to calculate P value of the specific region. Then P values are adjusted to Q values using Benjamini-Hochberg procedure [1]. Adjusted Q value is stricter than P value, which is more helpful to filter false positive peak. Q value <0.05 is the default threshold for MACS (version2.1.2), which was also used in previous study [2]. We have added the explanation in the revised manuscript (Line 400).
Q13. Line 432: The way of normalization using ef1a is described incorrectly in this sentence.
A13. Thanks for your kind advice. We have corrected the description in the revised manuscript (Line 451).
References
- Benjamini, Y., & Hochberg, Y. . Controlling the False Discovery Rate: a Practical and Powerful Approach to Multiple Testing. Journal of the Royal Statistical Society 1995, 57, 289-300.
- Ackermann, A.M.; Wang, Z.; Schug, J.; Naji, A.; Kaestner, K.H. Integration of ATAC-seq and RNA-seq identifies human alpha cell and beta cell signature genes. Mol Metab 2016, 5, 233-244, doi:10.1016/j.molmet.2016.01.002.
Reviewer 2 Report
Did the authors need to obtain ethical approval to use Orange-spotted groupers? if so include the approval number.
Fig.6 is very confusive. The authors need to group them properly along with the results/legends.
Section 4.1: The authors stated that they conducted experiments based on Wu et al., 2019. However, Wu et al 2019 did not detailed the experimental/treatment set-up.
Include the treatment schedule including the dose/duration/sampling day etc.
The authors need to include gonad histology after MT treatment.
The authors studied Cyp19 but not CYP11b.
Did the authors measured serum E2 and 11-KT level? This is important to check/determine the stability of sex reversal fate .
Authors also expected to include the TUNEL/PCNA staining, as the PCNA staining will reveal the germ cell origination.
Author Response
Dear reviewer,
Thank you for your comments concerning our manuscript entitled “Integration of ATAC-seq and RNA-seq unravels the chromatin accessibility during sex reversal in orange-spotted grouper (Epinephelus coioides)” (Manuscript Number: ijms-747820). Those comments are all valuable and very helpful for revising and improving our paper, as well as the important guiding significance to our research. We have studied comments carefully and have made correction that we hope meet with approval. Revised portion are marked in red in the paper. Regarding the questions (Q) raised by these reviewers, we would like to answer (A) them as follows.
Q1. Did the authors need to obtain ethical approval to use Orange-spotted groupers? if so include the approval number.
A1. Yes, all animal experiments were conducted in accordance with the guidelines and approval of the respective Animal Research and Ethics Committees of Sun Yat-Sen University. The approval file was attached here for your reference.
Q2. Fig.6 is very confusive. The authors need to group them properly along with the results/legends.
A2. Thanks for your kind advice. We have modified the description and legend of Fig.6 (Line 244-253, 266-268).
Q3. Section 4.1: The authors stated that they conducted experiments based on Wu et al., 2019. However, Wu et al 2019 did not detailed the experimental/treatment set-up.
A3. Thanks for your kind advice. We are sorry for the inconvenience, and we have added more description about the handling of animals and procedures in the revised manuscript (Line 366-381).
Q4. Include the treatment schedule including the dose/duration/sampling day etc.
A4. Thanks for your kind advice. We have supplemented the details of the treatment (Line 366-375), and listed the experimental design, sampling day, and duration in Table 4.
Q5. The authors need to include gonad histology after MT treatment.
A5. Thanks for your kind advice. We have added the gonad histology changes during sex reversal in figure 1 and detailed description (Line 77-93).
Q6. The authors studied Cyp19 but not CYP11b.
A6. Thanks for your kind advice. CYP19 and CYP11B are members of the cytochrome P450 (P450) superfamily. CYP19 is the terminal enzyme in the steroidogenic pathway and catalyzes the conversion of androgens to estrogens [1]. CYP11B catalyzes the biosynthesis glucocorticoids and mineralocorticoids in adrenal gland [2]. Given the important role in synthesizing hormone, we studied CYP19 but not CYP11b in this study. As the research moves along, we will study many enzymes in the production of hormones, including CYP11B, CYP21A, HSD11B and so on.
Q7. Did the authors measured serum E2 and 11-KT level? This is important to check/determine the stability of sex reversal fate.
A7. Thanks for your kind advice. 1 Our colleagues have measured serum E2 and 11-KT level during sex reversal many times [3,4] (Fig. 1). However, the hormone level affected by individual difference wasn’t always stable. So we use the histological morphology to identify the developmental stage which is the most direct and credible way. 2 Sex reversal induced by exogenous hormones is an effective method applied in many previous studies [5-7]. So far, it’s a quite stable way to obtain fish with differently developmental stage.
Fig. 1. Serum sex steroid hormone level during MT-induced sex reversal. (A) The E2 level in the control and MT-feeding group. (B) The 11-KT level in the control and MT-feeding group [4].
Q8. Authors also expected to include the TUNEL/PCNA staining, as the PCNA staining will reveal the germ cell origination.
A8. Thanks for your insightful advice. We have detected the proliferation and apoptosis in gonad during sex reversal. To investigate the proliferation signal in the gonad during MT-induced sex reversal, PCNA expression was examined using immunohistochemistry (IHC). In the control group, the PCNA signals were mainly located in somatic cells rather than oocytes (Fig.2a). At the early stage of sex reversal (one week after MT-implantation), PCNA-positive signals were observed in the newly developed male germ cells (Fig.2b). At the middle stages of sex reversal (three weeks after MT-implantation), PCNA signals were detected in the male germ cells (Fig.2c). At the late stages of sex reversal (five weeks after MT-implantation), the signals were only observed in spermatogonia and spermatocytes (Fig.2d).
Fig.2. The proliferation detection in the gonad during sex reversal. IHC was performed on the gonad in different stages with antibody against PCNA (red). The sections were counterstained with DAPI (blue), and the merged signals are purple. (a) Ovary in control group with the portion of mostly primary-growth stage oocytes (PO), b) gonad after one week by MT-implantation (the early stage of sex reversal), (c) gonad after three weeks by MT-implantation (the middle stage of sex reversal), (d) gonad after five weeks by MT-implantation (the late stage of sex reversal). PO, primary-growth stage oocyte; PVO, cortical-alveolus stage oocyte; SG, spermatogonia, SC, spermatocyte, ST, spermatid. Scale bars = 50 μm.
TdT-mediated dUTP nick end labeling (TUNEL) showed that The TUNEL signals can be hardly detected in the normal ovary (Fig.3a). At the early stage of sex reversal, TUNEL-positive signals could be found in some oocytes and somatic cells (Fig.3b). At the middle stage of sex reversal, TUNEL-positive signals were stronger and only observed in the oocytes (Fig.3c). At the late stage of sex reversal, the signals were in several cells which might be somatic cells or false positive signals (Fig.3d).
Fig.3. TUNEL assay in the gonads during sex reversal. TUNEL signals were green, and the sections were counterstained with DAPI (blue). (a) Ovary in sham group with the portion of mostly primary-growth stage oocytes, (b) gonad after one week by MT-implantation, (c) gonad after three weeks by MT-implantation, (d) gonad after five weeks by MT-implantation. PO, primary-growth stage oocyte; PVO, cortical-alveolus stage oocyte; SG, spermatogonia, SC, spermatocyte, ST, spermatid. Scale bars = 50 μm
The result of proliferation and apoptosis in gonad can provide us some information about the origination of male germ cells, which is a question we want to solve for a long time. However, we need lots of evidences to prove our hypothesis. I believe that we'll be able to answer the question clearly in future.
References
- Li, M.; Sun, L.; Wang, D. Roles of estrogens in fish sexual plasticity and sex differentiation. Gen Comp Endocrinol 2019, 277, 9-16, doi:10.1016/j.ygcen.2018.11.015.
- Schiffer, L.; Anderko, S.; Hannemann, F.; Eiden-Plach, A.; Bernhardt, R. The CYP11B subfamily. J Steroid Biochem Mol Biol 2015, 151, 38-51, doi:10.1016/j.jsbmb.2014.10.011.
- Wu, G.C.; Tey, W.G.; Li, H.W.; Chang, C.F. Sexual Fate Reprogramming in the Steroid-Induced Bi-Directional Sex Change in the Protogynous Orange-Spotted Grouper, Epinephelus coioides. PLoS One 2015, 10, e0145438, doi:10.1371/journal.pone.0145438.
- Wang, Q.; Liu, Y.; Peng, C.; Wang, X.; Xiao, L.; Wang, D.; Chen, J.; Zhang, H.; Zhao, H.; Li, S., et al. Molecular regulation of sex change induced by methyltestosterone -feeding and methyltestosterone -feeding withdrawal in the protogynous orange-spotted grouper. Biol Reprod 2017, 97, 324-333, doi:10.1093/biolre/iox085.
- Yeh, S.-L.; Kuo, C.-M.; Ting, Y.-Y.; Chang, C.-F. Androgens stimulate sex change in protogynous grouper, Epinephelus coioides: spawning performance in sex-changed males. Comparative Biochemistry and Physiology Part C: Toxicology & Pharmacology 2003, 135, 375-382, doi:10.1016/s1532-0456(03)00136-4.
- Shi, Y.; Zhang, Y.; Li, S.; Liu, Q.; Lu, D.; Liu, M.; Meng, Z.; Cheng, C.H.; Liu, X.; Lin, H. Molecular identification of the Kiss2/Kiss1ra system and its potential function during 17alpha-methyltestosterone-induced sex reversal in the orange-spotted grouper, Epinephelus coioides. Biol Reprod 2010, 83, 63-74, doi:10.1095/biolreprod.109.080044.
- Huang, M.; Wang, Q.; Chen, J.; Chen, H.; Xiao, L.; Zhao, M.; Zhang, H.; Li, S.; Liu, Y.; Zhang, Y., et al. The co-administration of estradiol/17alpha-methyltestosterone leads to male fate in the protogynous orange-spotted grouper, Epinephelus coioides. Biol Reprod 2019, 100, 745-756, doi:10.1093/biolre/ioy211.

Round 2
Reviewer 1 Report
Thank you for careful revisions.
Author Response
Dear reviewer,
Thank you for your comments concerning our manuscript entitled “Integration of ATAC-seq and RNA-seq unravels the chromatin accessibility during sex reversal in orange-spotted grouper (Epinephelus coioides)” (Manuscript Number: ijms-747820). We have checked the language carefully throughout the manuscript, and corrected many mistakes with red color in the revised manuscript. We hope you will be satisfied with our corrections.
Yours sincerely,
Xi Wu
School of Life Sciences,
Sun Yat-Sen University,
Guangzhou 510275, China
E-mail: [email protected]
Reviewer 2 Report
- The reply to reviewer, the authors stated that they included serum E2 and 11-KT level in Fig.1. But the manuscript do not have the data.
- Also the authors stated that Fig 2 is PCNA analysis through IHC. But the reviewer not able to find the PCNA results
- The authors stated that Fig 3a-3b have TUNEL assay results. Again the reviewer not able to find the TUNEL results
The authors need to include all the results mentioned in the reply to reviewer .
Author Response
Dear reviewer,
Thank you for your comments concerning our manuscript entitled “Integration of
ATAC seq and RNA seq unravels the chromatin accessibility during sex reversal in orange spotted grouper ( Epinephelus coioides )” (Manuscript Number ijms 747820).
We have studied comments carefully and have made correction that we hope meet with approval. Revised portion are marked in red in the paper. Regarding the questions (Q) raised by the reviewer , we would like to answer (A) them as follows.
Q1. The reply to reviewer, the authors stated that they included serum E2 and 11-KT level in Fig.1. But the manuscript do not have the data.
Also the authors stated that Fig 2 is PCNA analysis through IHC. But the reviewer not able to find the PCNA results
The authors stated that Fig 3a-3b have TUNEL assay results. Again the reviewer not able to find the TUNEL results
The authors need to include all the results mentioned in the reply to reviewer.
A1. Thank you for your kind advice. We have showed the Fig.1-3 in the response to reviewer report, however, we are so sorry that you couldn’t find the figures. We guess it’s the bug of the submission system. So we also attached a cover letter including figures for your better understanding.
“Fig.” was used in the response in order to distinguish the “Figure” in the manuscript. Depend on our experience, the level of hormone in serum can help us to identify the developmental stage of gonad, but it isn’t always stable for
fish. So we use the histological morphology to identify the developmental stage
which i s the most direct and credible way.
At the beginning, we tried to utilize the proliferation and apoptosis in
gonad to identify the origination of new male germ cells, however, we hadn’t
enough proofs to prove our hypothesishypothesis. Thanks for your insightfulinsightful adviceadvice, and our colleagues will our colleagues will continue to push the project. Now, we show the results of results of PCNA analysis and TUNEL assay for reference. And we think the results is irrelevant with the topic of this study, so we didn’t put them in the revised irrelevant with the topic of this study, so we didn’t put them in the revised manuscript.
Below is the the response in round 1 including including Fig.1-3.
Round 1
Q7. Did the authors measured serum E2 and 11-KT level? This is important to check/determine the stability of sex reversal fate.
A7 . Thanks for your kind advice.
1 Our colleagues have measured serum E2 and 11 KT level during sex reversal many times Fig. 1) However, the hormone level affected by individual difference wasn’t always stable . So we use the histological morphology to identify the developmental stage which i s the most direct and credible way.
2 Sex reversal induced by exogenous hormones is an effective method applied in many previous studies [3 5] So far, it’s a qui te stable way to obtain fish with differently developmental stage.
Fig.1 (attached)
Q8. Authors also expected to include the TUNEL/PCNA staining, as the PCNA staining will reveal the germ cell origination.A8. Thanks for your insightful advice. We have detected the proliferation and apoptosis in gonad during sex reversal. To investigate the proliferation signal in the gonad during MT induced sex reversal, PCNA expression was examined using immunohistochemistry (IHC). In the control group, the PCNA signals were mainly located in somatic cells rather than oocytes (Fig.2a). At the early stage of sex reversal (one week after MT implantation), PCNA positive signals were observed in the newly developed male germ cells (Fig. 2b ). At the middle stages of sex reversal (three weeks after MT implantation), PCNA signals were detected in the male germ cells ( 2c ). At the late stages of sex reversal (five weeks after MT implantation), the signals were only observed in spermatogonia and spermatocytes (Fig. 2d ).
Fig.2 (attached)
TdT mediated dUTP nick end labeling (TUNEL) showed that The TUNEL signals
can be hardly detected in the normal ovary (Fig.3a). At the early stage of sex reversal, TUNEL positive sig nals could be found in some oocytes and somatic cells (Fig. 3b). At the middle stage of sex reversal, TUNEL positive signals were stronger and only observed in the oocytes (Fig. 3c). At the late stage of sex reversal, the signals were in several cells which might be somatic cells or false positive signals (Fig.3d)
Fig.3 (attached)
The result of proliferation and apoptosis in gonad can provide us some information about the origination of male germ cells which is a question we want to solve for a long time . However, we need lots of evidences to prove our hypothesis I believe that we'll be able to answer th e question clearly in future.
References
1.Wu, G.C.; Tey, W.G.; Li, H.W.; Chang, C.F. Sexual Fate Reprogramming in the Steroid Induced Bi Directional Sex Change in the Protogynous Orange Spotted Grouper, Epinephelus coioides . PLoS One 2015 , 10 , e0145438, doi:10.1371/journal.pone.
2. Wang, Q.; Liu, Y.; Peng, C.; Wang, X.; Xiao, L.; Wang, D.; Chen, J.; Zhang, H.; Zhao, H.; Li, S., et al. Molecular regulation of sex change induced by methyltestosterone feeding and methyltestosterone feeding withdrawal in the protogynous orange spotted grouper. Biol Reprod 2017 , 97 , 324 333, doi:10.1093/biolre/iox085.
3. Yeh, S. L.; Kuo, C. M.; Ting, Y. Y.; Chang, C. F. Androgens stimulate sex change in protogynous grouper, Epinephelus coioides : spawning performance in sex changed males. Comparative Bioc hemistry and Physiology Part C: Toxicology & Pharmacology 2003 , 135 , 375 382, doi:10.1016/s1532 0456(03)00136 4.
4. Shi, Y.; Zhang, Y.; Li, S.; Liu, Q.; Lu, D.; Liu, M.; Meng, Z.; Cheng, C.H.; Liu, X.; Lin, H. Molecular identification of the Kiss2/Kiss1ra system and its potential function during 17alpha methyltestosterone induced sex reversal in the orange spotted grouper, Epinephelus coioides . Biol Reprod 2010 , 83 , 63 74, doi:10.1095/biolreprod.109.080044.
5. Huang, M.; Wang, Q.; Chen, J.; Chen, H.; Xiao, L.; Zhao, M.; Zhang, H.; Li, S.; Liu, Y.; Zhang, Y., et al. The co administration of estradiol/17alpha methyltestosterone leads to male fate in the protogynous orange spotted grouper, Epinephelus coioides . Biol Reprod 2019 , 100 , 745 756, doi:10.1093/biolre /
Yours sincerely.

Round 3
Reviewer 2 Report
The authors stated that they included serum E2 and 11-KT level (Fig.1), PCNA (Fig.2), TUNEL (Fig.3). However the revised manuscript do not have the data. The authors have the image in the cover letter, but the pictures needs to be included in the revised manuscript.
In the revised manuscript, Fig.1 is Gonadal histological morphology;
Fig.2 is ATAC-seq results in the gonad;
Fig 3 is TAC-seq and RNA-seq profiles
The authors need to make sure that they include the results/graphs/images for E2, 11-KT, PCNA, TUNEL in the revised manuscript.
Author Response
Dear reviewer,
Thank you for your comments concerning our manuscript entitled “Integration of ATAC-seq and RNA-seq unravels the chromatin accessibility during sex reversal in orange-spotted grouper (Epinephelus coioides)” (Manuscript Number: ijms-747820). We have studied comments carefully and have made correction that we hope meet with approval. Revised portion are marked in red in the paper. Regarding the questions (Q) raised by the reviewer, we would like to answer (A) them as follows.
Q1. The authors stated that they included serum E2 and 11-KT level (Fig.1), PCNA (Fig.2), TUNEL (Fig.3). However the revised manuscript do not have the data. The authors have the image in the cover letter, but the pictures needs to be included in the revised manuscript.
In the revised manuscript, Fig.1 is Gonadal histological morphology;
Fig.2 is ATAC-seq results in the gonad;
Fig 3 is TAC-seq and RNA-seq profiles
The authors need to make sure that they include the results/graphs/images for E2, 11-KT, PCNA, TUNEL in the revised manuscript.
A1. Thanks for your kind advice. Our colleagues have measured serum E2 and 11-KT level during sex reversal many times [1,2]. Herein, we didn’t put the result in the manuscript. Besides, we have added the results of PCNA and TUNEL (Figure 2-3) and the detailed description in the revised manuscript (Line 95-125, 333-341, 421-436). We are sorry for the inconvenience.
References
- Wu, G.C.; Tey, W.G.; Li, H.W.; Chang, C.F. Sexual Fate Reprogramming in the Steroid-Induced Bi-Directional Sex Change in the Protogynous Orange-Spotted Grouper, Epinephelus coioides. PLoS One 2015, 10, e0145438, doi:10.1371/journal.pone.0145438.
- Wang, Q.; Liu, Y.; Peng, C.; Wang, X.; Xiao, L.; Wang, D.; Chen, J.; Zhang, H.; Zhao, H.; Li, S., et al. Molecular regulation of sex change induced by methyltestosterone -feeding and methyltestosterone -feeding withdrawal in the protogynous orange-spotted grouper. Biol Reprod 2017, 97, 324-333, doi:10.1093/biolre/iox085.
Yours sincerely,
Xi Wu
School of Life Sciences,
Sun Yat-Sen University,
Guangzhou 510275, China
E-mail: [email protected]